# Effectiveness of Plyometric and Eccentric Exercise for Jumping and Stability in Female Soccer Players—A Single-Blind, Randomized Controlled Pilot Study

**DOI:** 10.3390/ijerph18010294

**Published:** 2021-01-03

**Authors:** Guillermo Porrati-Paladino, Rubén Cuesta-Barriuso

**Affiliations:** Department of Physiotherapy, Faculty of Sport Sciences, European University of Madrid, 28670 Madrid, Spain; g.porratipaladino@gmail.com

**Keywords:** stability, jumping, plyometry, eccentric, soccer player, randomized clinical trial

## Abstract

Hamstring muscle injury is common in female soccer players. Changes affecting eccentric strength, flexibility, and the quadriceps–hamstring contraction cycle are risk factors associated with this type of injury. **Methods**: Seventeen soccer players were randomized to two groups: experimental (plyometric and eccentric exercises without external loads) and control (eccentric exercises without external loads). Eighteen sessions were scheduled over 6 weeks. The exercise program included three plyometric exercises (single-leg squat and lunge, 180 jump, and broad jump stick landing) and three eccentric exercises (Nordic hamstring exercise, diver, and glider). Dependent variables were jumping height (My Jump 2.0 App) and anterior, posteromedial, and posterolateral lower limb stability (Y-Balance test). **Results**: Following intervention, improvements were found in anterior and posteromedial stability (*p* = 0.04) in the experimental group. Posterolateral stability improved in athletes included in the control group (*p* = 0.02). There were differences in the repeated measures analysis for all variables, with no changes in group interaction (*p* > 0.05). **Conclusions**: Eccentric exercises, either combined with plyometric exercises or alone, can improve lower limb stability. No changes in jump height were noted in either group. There were no differences between the two groups in the variables studied. Future studies should analyze the effect of external loads on jumping stability and height in the performance of plyometric exercises.

## 1. Introduction

Female soccer players experience longer absences due to injury than men, both in those playing on artificial and natural turf, as well as during training and competition [1,2]. The body region subject to the highest injury rate in female sport is the lower limb [2,3], with the thigh being one of the most affected parts, along with the knee and ankle joints [2,4,5]. Furthermore, overuse injuries are more common than direct trauma injuries [6,7].

Injury prevention protocols have shown reduced efficacy [4,8]. In female sports, the incidence of hamstring injury is significantly high [9,10]. This injury is associated with pain in the posterior thigh region and the structural alteration of muscle fibers [11]. The main etiological factors for the development of hamstring injuries include poor hamstring–quadriceps ratio, the recurrence of previous injuries, muscle fatigue, and reduced hamstring flexibility [8,12,13,14].

The use of eccentric exercise protocol(s) has shown its effectiveness in reducing the incidence and recovery time of hamstring muscle injury [15]. For their part, plyometric exercises have shown to be effective in improving lower limb joint stability and coactivation of the hamstrings and quadriceps in female soccer players [16,17].

Moreover, performing plyometric exercises can improve the ability to endure prolonged and high-intensity exertion in both males and females [18]. The kinetic characteristics of the landing phase in plyometric exercises can be quantified to establish exercise intensity and its progression in a training program. Landing characteristics in plyometric exercises can be quantified to measure dynamic postural stability. The performance of plyometric exercises should progress from a shortest stabilization time at the beginning of the program, increasing the intensity with a similar increase in stabilization time [19].

The aim of this study was to verify the effectiveness of a plyometric training program combined with eccentric exercises, when compared to eccentric training alone, in improving lower limb stability and jumping ability in federated female soccer players between 18 and 30 years of age.

## 2. Materials and Methods

### 2.1. Ethical Approval

This study was a randomized, single-blind controlled pilot study with follow-up. This study was approved by the Research Committee of the European University of Madrid (registration No.: CIPI/18/030). The players who agreed to participate in the study signed an informed consent document, which was drafted in accordance with the Helsinki Declaration. This study was registered in an international registry (ClinicalTrials.gov ID: NCT04255290).

### 2.2. Participants

A representative sample of the sample under study was calculated. The magnitude of this difference was considered by calculating the effect size (d = 0.43) [16] for measuring the vertical jump in university soccer players. With an alpha level (type I error) of 0.05, a statistical power of 80% (1−β = 0.80), and a non-sphericity correction of 1, a sample size of 68 soccer players was estimated. As a randomized pilot study, 17 soccer players were recruited. The calculation was performed using G*Power software, version 3.1.9.4. (Heinrich-Heine-Universität, Düsseldorf, Germany)

The players were recruited from the sports centers Las Rozas S.A.D., Las Rozas female’s club, and Electrocor C.F between February and May 2019. The study included athletes who met the following criteria: female; 18 to 30 years old; trained up to 3 days per week; and federated in the community of Madrid (Spain). Exclusion criteria were players who had an injury at the time of starting the study or could not follow the protocol designed for the intervention; those with a history of hamstring injury in the 6 months prior to the study; those who played another federated sport; and those who failed to sign the informed consent document.

Once the informed consent document was signed, and after checking that all subjects met the selection criteria, they were randomized to the two study groups: experimental and control. Randomization was performed by a person unrelated to the study using the opaque envelope system. Subjects assigned to the experimental group (n = 9) received an intervention using plyometric and eccentric hamstring exercises, and those in the control group (n = 8) received an intervention using eccentric hamstring exercises.

### 2.3. Instruments

Three evaluations were performed: pretreatment (T0), post-treatment (T1), and at 4 weeks follow-up (T2). All evaluations were carried out by the same physiotherapist, blinded to the players’ allocation to the study groups, under the same conditions and following the same evaluation protocol. The dependent variables of this study were jumping and lower limb stability. The measurement instruments used to evaluate dependent variables were as follows:-Application “My jump 2.0”. With this mobile app, jumps were evaluated [20]. The rater stood at a distance of 1.5 m from the player, and a smartphone was placed at ground level to register the measurements. The test consisted of countermovement jump, starting from two-legged stance with hands on hips. Then, the athlete performed a jump from a 90º knee flexion position (avoiding bending the trunk). During the flight phase, the legs should be extended. When contacting the platform, the feet rested first on the metatarsal and subsequently on the back region. The unit of measurement for this instrument is centimeters.-Y-Balance test. This instrument was used to assess lower limb stability in accordance with the protocol developed by Plisky et al. [21]. The Y-Balance test is a validated derivation of the Star Excursion Balance Test (SEBT). This instrument uses the anterior, posteromedial, and posterolateral components of the SEBT to evaluate neuromuscular characteristics such as lower extremity coordination, balance, flexibility, and strength. The subject stood in monopodal stance in the center of an inverted Y-shape on the leg to be evaluated. Three attempts were made for each reaching direction (anterior, posteromedial, and posterolateral). The distance was measured in centimeters, and the arithmetic mean of three attempts made in each range was calculated [22]. The Y-shape was made with tape on the floor. Distance was measured with a tape measure.

Before starting the intervention, the main independent sociodemographic variables (age), anthropometric variables (weight, height, and body mass index), and sports variables (dominance, years federated, matches played this season, and position in the field) were measured.

Prior to pretreatment assessment, a pilot trial was conducted for the rater and the main researcher (with 5 subjects) and with the aim of calculating the interobserver reliability in the measurement of dependent variables. A medium-high value was observed in the interrater reliability analysis for jumping (ICC = 0.72) and lower limb stability (ICC = 0.69) variables.

### 2.4. Experimental Design

Three weekly sessions were held over a period of 6 weeks. Each session lasted 20 min in the experimental group and 12 min in the control group. All interventions were performed at the beginning of the training session and were supervised by a physiotherapist with experience in sports physiotherapy. The intervention based on eccentric exercise was performed in pairs with three exercises: Nordic hamstring exercise, diver, and glider. For all eccentric exercises, 2 sets of 5 repetitions were performed, with 20 s rest between sets. The intensity was progressively increased until reaching 3 sets with 10 repetitions and 30 s rest in the last two weeks.

-The Nordic hamstring exercise was performed using the protocol designed by Van der Horst et al. [23]. One subject in each pair kneeled, while the other subject, behind her, held her legs. The first one let herself drop forward in a controlled manner until touching the ground.-To perform “the diver” exercise, we followed the protocol described by Askling et al. [24]. The player stood in monopodal stance performing a hip flexion while maintaining support. She was asked to bring her arms forward, while moving the contralateral lower limb backwards. The knee should be supported at around 10–20 degrees flexion and the exercise, was performed slowly, returning to the starting position.-“The glider” exercise was conducted according to the protocol described by Askling et al. [24]. The athlete stood on both feet in front of her partner while holding each other’s shoulders and gliding one leg backwards while the other remained steady. She then returned to the starting position with the help of her partner, without letting the knee fall below 10 degrees flexion.

For all eccentric exercises, 2 sets of 5 repetitions were performed, with 20 s rest between sets. The intensity was progressively increased until completing 3 sets with 10 repetitions and 30 s rest between sets.

The intervention with plyometry exercises was performed adapting the protocol designed by Tsang and Di Pasquale [25], which included single-leg squat and lunge, 180 jump and broad jump stick landing exercises.

-For the single-leg squat and lunge exercise, the athlete adopted a squatting position with her partner behind her, lifting and holding the back leg. The player then performed a monopodal jump with the supported leg. In the first week, a set with 5 repetitions and 30 s rest between sets was performed, increasing the intensity until reaching 3 sets with 8 repetitions in the last two weeks of the study.-To perform the 180 jump exercise, the player started from a bipedal stance position, with the trunk upright and hands on hips. She was asked to jump with both legs while turning 180 degrees in rotation during the jump, attempting to sustain the fall for 2 s. In each repetition, rotation was performed in a different direction. Two 20 s sets were performed, with 20 s rest between sets. Every two weeks, the intensity increased until reaching 3 sets of 30 s in the last two weeks.-For the broad jump stick landing exercise, the players stood in bipedal support, with their hands free, jumping with both feet as far as possible. The knees should not go beyond the tips of the toes and the fall should be with the trunk as straight as possible. Five jumps were performed in the first two weeks, increasing the number to 8 in the next two weeks, and eventually reaching 10 jumps in the last two weeks of the study.

### 2.5. Statistical Analysis

Statistical analysis was performed using SPSS (version 25.0) (IBM Corp. Released 2017. IBM SPSS Statistics for Windows. Armonk, NY) for Windows. The main descriptive statistics of the independent variables (mean and standard deviation) were calculated. Homogeneity in sample distribution between the two groups was calculated using the Shapiro–Wilk test. Interobserver reliability was calculated using the intraclass correlation coefficient.

The changes between the different evaluations were observed using Student’s parametric T test for paired samples. The ANOVA test of repeated measures provided the intra-subject effect and group interaction. The error rate of the significance level was controlled by Bonferroni correction. When the Mauchly sphericity test was significant, the Greenhouse–Geisser correction coefficient was used. The partial eta-squared value was calculated as an indicator of effect size (classified as small (0.01), medium (0.06), or large (0.14)) [26]. An intent-to-treat analysis was performed in this study. Differences between groups were considered statistically significant for *p* < 0.05.

## 3. Results

Of the 17 subjects participating at the start the study, two abandoned during the experimental phase due to patella dislocation and grade 2 ankle sprain. Both injuries were unrelated to the intervention. During follow-up, another athlete left the study due to timetable incompatibility problems. As such, the study was ultimately completed by 15 players (8 in the experimental group and 7 in the control group). At baseline, both groups were homogeneous (*p* > 0.05) in all dependent variables except for anterior right stability (*p* = 0.03). Figure 1 shows the flow diagram of the study. During the study, none of the players took hormonal birth control, oral contraception, or indicated the use of intrauterine device (IUDs). At the beginning of the study, 10 women were in the follicular phase and 7 in the luteal phase. After allocation to the study groups, there were no statistical differences between the groups regarding menstrual cycle status. Table 1 shows the main descriptive statistics of the soccer players included in the study. Table 2 shows descriptive statistics of the study variables at posttreatment and follow-up assessments. 

Subsequent to intervention, the players included in the experimental group exhibited changes in anterior stability of the left and right leg (*p* = 0.04) and posteromedial stability of the right leg (*p* = 0.04). When comparing evaluations at T0–T2, changes were found in the anterior (*p* = 0.04) and posteromedial stability of the left leg (*p* = 0.04). Regarding the control group, we found changes in the posterolateral stability of the right leg (*p* = 0.02) after the intervention. Significant changes (*p* < 0.05) were found in all variables after follow-up, except for jumping (*p* = 0.96). No changes (*p* > 0.05) were found at the T1–T2 comparison in either group. Table 3 shows the mean difference of all dependent variables, together with significant changes when comparing the assessments.

Repeated measures analysis revealed significant differences in all variables (*p* < 0.05) depending on the time evaluated. When calculating the partial eta-squared value, high effect size values were found for all measured variables (η^2^_p_ > 0.14). However, no group interaction changes were revealed (*p* > 0.05). Table 4 shows the analysis of repeated measures and intergroup interaction.

Pairwise comparison analysis revealed significant changes (*p* < 0.05) in the T0–T1 analysis for anterior stability of the left leg (*p* = 0.01), and posteromedial (*p* = 0.03) and posterolateral (0.02) stability of the right leg. No changes (*p* > 0.05) were found in the analysis comparing T1–T2 measurements. There were significant changes between T0–T2 measurements in all variables, except for jumping (*p* = 0.06) and posterolateral stability of the left leg (*p* = 0.09). Table 5 shows the pairwise comparison analysis.

## 4. Discussion

After the intervention and follow-up period, we observed improved lower limb stability in athletes of both study groups. However, we found no changes in jump performance in any of the evaluations. When comparing changes between the two groups, no inter-group differences were noted in any of the dependent variables. Thus, the study hypothesis was not confirmed, thus accepting the alternative hypothesis.

After the intervention, improved stability was observed in the two study groups. In terms of lower limb stability values, different studies have reported improvements in stability in basketball [27] and soccer [28] players using a single stability parameter. Benis et al. [29] conducted a protocol involving body-weight neuromuscular exercises, observing improvements in posteromedial and posterolateral measurements using the Y-Balance test. Their study, using an exercise intensity of 16 sessions over an 8-week period, found improvements in anterior knee stability.

The athletes included in the two study groups performed eccentric exercises. In both groups we observed an improvement in lower limb stability. Muscle strength and architectural characteristics are adaptable and can be modified by a number of stimuli, including eccentric strength training [30]. Eccentric training of the hamstrings has been shown to produce neuromuscular adaptations. These adaptations include an increase in the length of the fascicle of the long head of the biceps femoris muscle [31], an increase in muscle strength or volume [31,32,33], and an increase in the ability to generate higher torque levels at longer muscle lengths [32].

The experimental group performed a combined protocol of plyometric and eccentric exercises. Plyometric exercises using the athlete’s body weight train the muscles, connective tissue, and nervous system to effectively carry out the stretch-shortening cycle [29]. Plyometric exercises can improve neuromuscular control in female athletes [34].

In terms of jump assessment, we found no change in either study group. Ramirez-Campillo et al. [18] compared the differences between combining the plyometric training with load-bearing or non-load-bearing exercise, observing improvements in the group that performed load-bearing exercises. Performing non-load-bearing exercises may be involved in the absence of changes in jumping-based exercises on hamstring–quadriceps muscle contraction. This hypothesis would be consistent with the absence of changes found in both groups of our study.

Although the sample size was small, high values were reported in the effect size in the analysis of the intra-group effect. The absence of significant differences in group interaction indicated the improvement in the variables was not dependent on athlete allocation to the study groups. We can, thus, establish that both interventions were effective in improving lower limb stability and jumping performance in soccer players, without any differences between other exercise protocols. An improvement in performance in the results obtained using the Y-Balance test measuring instrument is associated with a lower risk of lower limb injury in non-contact sports [35]. In addition, a positive correlation has been observed [36] between lower limb strength and range in the different directions in the Y-Balance test in women. Although the follow-up evaluation in our study was not intended to assess the effectiveness in injury prevention, the changes observed in the measurement of lower limb stability suggest that performance in the Y-Balance test may serve as an indication to assess the risks of lower limb injury in study athletes.

### 4.1. Study Limitations

The small sample size was the main limitation of this study. However, we compensated for this limitation by implementing several methodological quality-control measures (blinding of the rater, interobserver reliability analysis, follow-up assessment, etc.).

Similarly, the quadriceps–hamstring muscle contraction ratio was not measured directly, which could have yielded more exact and objective results; however, a more specific device was not available due to the high expense involved. One factor that may have limited the results was the drop-out rate during the study for reasons unrelated to the research. We attempted to address and compensate for this limitation by applying an intent-to-treat analysis.

### 4.2. Recommendations for Clinical Practice

Both exercise programs can help in the prevention of hamstring muscle injury. Its progressive periodization, the reduced time needed to perform it, and the simplicity of the exercises make it a feasible option for therapeutic sports injury prevention programs. Because the exercises can be carried out without the need for additional aids or appliances, these can be easily implemented by a therapeutic work team.

### 4.3. Recommendations for Future Research

Randomized clinical trials are needed to confirm the findings of this study. It is recommendable to use more objective measuring instruments, a larger sample size of soccer players, and multicenter recruitment. Lastly, it is advisable to carry out the plyometric exercises under load-bearing conditions.

Conducting medium- and long-term studies could promote the effectiveness of these interventions for injury prevention in soccer players. The implementation of a longer follow-up period could confirm the suitability of these exercise programs for preventive purposes.

## 5. Conclusions

An eccentric exercise program, combined with plyometric exercises, or alone, can improve lower limb dynamic stability in female football players. Neither intervention improved jump height. There were no differences between the two groups in the interaction analysis. Future studies should analyze the effect of weight implementation in plyometric exercises with a larger recruitment of athletes.

## Figures and Tables

**Figure 1 ijerph-18-00294-f001:**
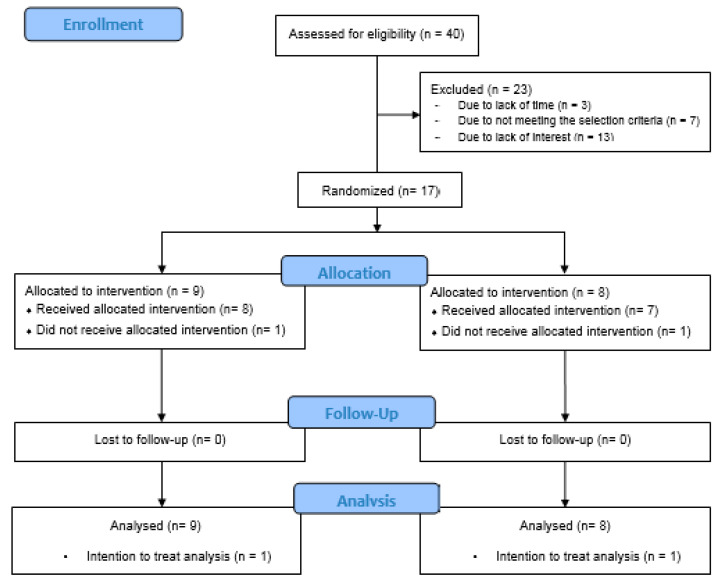
Study flow diagram.

**Table 1 ijerph-18-00294-t001:** Descriptive analysis (mean and standard deviation) at baseline of the total sample, and according to study group.

Variables	All Sample	Experimental Group	Control Group	*p* Value ª
Age (years)	21.71 (3.54)	21.11 (4.16)	22.38 (2.82)	0.06
Height (cm.)	162.71 (4.37)	163 (4.63)	162.38 (4.34)	0.33
Weight (kg) *	63.87 (12.51)	61.83 (9.17)	66.16 (15.81)	0.03
Body mass index (kg/m^2^) *	24.02 (4.70)	23.07 (3.41)	25.07 (5.90)	0.03
Federated years (years)	7.24 (5.94)	7.56 (7.56)	6.87 (3.87)	0.08
Matches played in the season (number)	10.82 (4.90)	10.44 (5.36)	11.25 (4.65)	0.06
	n (%)	n (%)	n (%)	
Menstrual cycle status (follicular phase/luteal phase)	10/7	6/4	4/3	0.07 ^b^
Jumping height (cm)	47.08 (1.65)	45.76 (5.18)	48.72 (14.83)	0.05
Right anterior stability (cm) *	57.88 (4.41)	60.22 (3.66)	55.25 (3.77)	0.03
Right posteromedial stability (cm)	61.00 (9.10)	64.22 (7.22)	57.38 (10.07)	0.72
Right posterolateral stability (cm)	63.65 (9.37)	64.11 (10.77)	63.13 (8.21)	0.50
Left anterior stability (cm)	57.47 (4.75)	59.67 (4.44)	55.00 (4.00)	0.95
Left posteromedial stability (cm)	57.53 (6.61)	59.22 (6.20)	55.63 (6.94)	0.93
Left posterolateral stability (cm)	68.06 (8.47)	70.44 (9.20)	65.38 (7.19)	0.35

^a^ Shapiro–Wilk test; ^b^ Fisher exact test; * Significant difference (*p* < 0.05).

**Table 2 ijerph-18-00294-t002:** Descriptive statistics (mean and standard deviation) of the study variables at posttreatment and follow-up assessments.

Variables	Experimental Group		Control Group	
	T1	T2	T1	T2
Jumping height (cm)	46.14 (6.05)	45.55 (6.03)	50.18 (13.65)	49.43 (11.94)
Right anterior stability (cm)	63.11 (4.28)	63.11 (4.67)	59.38 (7.46)	61.25 (5.23)
Right posteromedial stability (cm)	69.67 (6.32)	67.56 (5.48)	65.00 (3.92)	63.88 (7.66)
Right posterolateral stability (cm)	69.78 (11.91)	68.56 (10.94)	70.38 (5.39)	71.13 (5.38)
Left anterior stability (cm)	62.44 (3.39)	62.78 (4.14)	57.88 (4.19)	58.38 (4.68)
Left posteromedial stability (cm)	62.33 (4.97)	64.56 (7.09)	62.00 (7.54)	64.25 (6.49)
Left posterolateral stability (cm)	73.11 (8.47)	71.11 (8.23)	66.88 (5.16)	69.88 (6.70)

Outcome measures after the 3-week period of experimental and control interventions (T1) and after a further 4-week follow-up (T2).

**Table 3 ijerph-18-00294-t003:** Statistical analysis of the difference in means (and statistical significance) of the study variables between the baseline and post-treatment, and baseline and follow-up assessments.

Variables	Measure Instrument	Experimental Group			Control Group		
		T0–T1	T1–T2	T0–T2	T0–T1	T1–T2	T0–T2
Jump	Jumping height	−0.44 (0.16)	−0.13 (0.61)	0.30 (0.07)	−0.26 (0.10)	−0.27 (0.10)	−0.01 (0.96)
Lower limb stability	Right anterior	−2.88 (0.04) *	0.00 (1.00)	−2.88 (0.13)	−4.12 (0.15)	−1.87 (0.13)	−6.0 (0.01) *
	Right posteromedial	−5.44 (0.03) *	2.11 (0.14)	−3.33 (0.12)	−7.62 (0.10)	1.12 (0.69)	−6.5 (0.01) *
	Right posterolateral	−5.66 (0.11)	1.22 (0.26)	−4.44 (0.09)	−7.25 (0.02) *	−0.75 (0.66)	−8.0 (0.04) *
	Left anterior	−2.77 (0.04) *	−0.33 (0.71)	−3.11 (0.04) *	−2.87 (0.07)	−0.50 (0.69)	−3.3 (0.02) *
	Left posteromedial	−3.11 (0.17)	−2.22 (0.05)	−5.33 (0.03) *	−6.37 (0.10)	−2.25 (0.17)	−8.62 (0.02) *
	Left posterolateral	−2.66 (0.26)	2.00 (0.19)	−0.66 (0.65)	−1.5 (0.45)	−3.00 (0.07)	−4.5 (0.02) *

MD: mean difference; T0–T1: outcome measures between baseline to posttreatment assessments; T1–T2: outcome measures between posttreatment to follow-up assessments; T0–T2: outcome measures between baseline to follow-up assessments. * Significant difference between improvements in the study groups (*p* < 0.01).

**Table 4 ijerph-18-00294-t004:** Within-subject and group interaction results in each of the dependent variables of the study and for the study groups.

Variable	Measure	Mauchly Sphericity		Intra-Group Effect			Inter-Group Interaction		
		W	Sig.	F	Sig.	η^2^_p_	F	Sig.	η^2^_p_
Jump	Jumping height	0.93	0.62	0.97	0.04 *	0.16	0.53	0.59	0.03
Lower limb stability	Right anterior ^a^	0.62	0.03	6.77	0.00 *	0.31	0.93	0.38	0.05
	Right posteromedial ^a^	0.58	0.02	5.29	0.02 *	0.26	0.43	0.58	0.02
	Right posterolateral ^a^	0.49	0.00	5.96	0.01 *	0.28	0.51	0.53	0.03
	Left anterior	0.94	0.65	4.44	0.02 *	0.22	0.01	0.98	0.00
	Left posteromedial ^a^	0.48	0.00	7.17	0.01 *	0.32	0.68	0.45	0.04
	Left posterolateral	0.72	0.10	3.46	0.04 *	0.18	2.38	0.10	0.13

W: Mauchly Sphericity Test; Sig.: significance. η^2^_p_: partial eta-squared. ^a^ The df corresponds to Greenhouse–Geisser test; * Significant difference (*p* < 0.05).

**Table 5 ijerph-18-00294-t005:** Pairwise comparison analysis, mean difference (and 95% confidence interval) between the three evaluations performed in each study group.

Variable	Measure	T0–T1			T1–T2			T0–T2		
		I-J	95% CI	Sig.	I-J	95% CI	Sig.	I-J	95% CI	Sig.
Jump	Jumping height	−0.35	−0.80, 0.19	0.14	0.15	−0.20, 0.51	0.76	−0.20	−0.62, 0.41	0.06
Lower limb stability	Right anterior	−3.47	−7.12, 0.18	0.06	−0.88	−2.82, 1.06	0.72	−4.35	−7.72, −0.98	0.01 *
	Right posteromedial	−6.47	−12.46, −0.47	0.03 *	1.64	−2.35, 5.64	0.85	−4.82	−8.51, −1.12	0.00 *
	Right posterolateral	−6.41	−12.05, −0.76	0.02 *	0.29	−2.25, 2.84	1.00	−6.11	−11.46, −0.76	0.02 *
	Left anterior	−2.82	−5.18, −0.45	0.01 *	−0.41	−2.37, 1.55	1.00	−3.23	−5.64, −0.82	0.00 *
	Left posteromedial	−4.64	−9.38, 0.54	0.08	−2.23	−4.56. 0.09	0.06	−6.88	−11.85, −1.90	0.00 *
	Left posterolateral	−2.11	−6.09, 1.86	0.51	−0.35	−3.06, 2.35	1.00	−2.47	−5.28, 0.34	0.09

T0–T1: outcome measures for baseline to posttreatment assessments, T1–T2: outcome measures for posttreatment to follow-up assessments: T0–T2: outcome measures for baseline to follow-up assessments; I-J: mean difference; 95%CI: 95% confidence interval; * Significant difference (*p* < 0.05).

## Data Availability

Not applicable.

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
