# Peer review of "Effectiveness of Plyometric and Eccentric Exercise for Jumping and Stability in Female Soccer Players—A Single-Blind, Randomized Controlled Pilot Study"

_ijerph, 2021, doi:10.3390/ijerph18010294_

Round 1

Reviewer 1 Report

Dear authors, the paper was clear and well written, however some considerations are suggested:

Please consider revising your title, (1) I do not think you have measured hamstring strength; (2) make it clear what outcome measure is the "jumping" reference (eg., Performance, hight,...) (3) the Y balance test has been referred as a test of dynamic postural control and stability, so perhaps you should elaborate a little bit more here and throughout the document; (4) why don’t you refer to your study as a single-blind, randomized controlled pilot study?

In the abstract section please consider changing the sentence on line 14 from “factors related to this injury” to “factors for this injury”. In the abstract you mention 17, but in the methods section you mention 14 subjects (7 for each group), please make it more clear. In relation to your eccentric exercises, do you consider them to be calisthenics? If so, please add the information that no external weight was added. You did not mention anywhere else in the document this “rupture angle” and again, I don’t really think you may say you have measured hamstring strength (line 18, 20, 23, 24). In the reference to the single-leg squat and lunge (line 16 and 17), please make it clear this is a plyometric training. In the conclusions section, in the abstract (line 24-26), It is not possible to conclude that one intervention is better than the other without comparing them statistically and I do not think you have added these results to the document.

In the introduction section, please consider adding “furthermore” before overuse in line 33. Please revise the English language and style for the sentence “have contributed few provide evidence considered to be sufficiently effective” on line 35/36. Line 38-40, in the sentence “The main etiological…. Flexibility” please consider the following “The main etiological factors for the development of hamstring injuries are poor hamstring-quadriceps ratio, the recurrence of previous injuries…”, please make it clear what you mean by “fatigue in the stages of greater muscle exhaustion”. Line 46, in the sentence “the aim of this study is to verify the effectiveness of plyometric and eccentric exercises in improving the lower limb stability…” please consider the following “The aim of this study is to verify the effectiveness of a plyometric training program combined with eccentric exercises when compared to eccentric training alone in ….”

Please organize the materials and methods section in subheadings, namely ethical approval, participants, instruments, experimental design, and statistical analysis. Line 50, why don’t you add controlled after the single-blind statement? Line 70, please remove the word “objectives”. Line 77, please make a reference to the outcome measure related to the word "jumping". Line 97, why did you calculate interobserver reliability? Did both the rater and the researcher collect the data? And where are these results and the results for the intraobserver analysis, that is perhaps even more important? Line 140-143, You refer to the use of the median and interquartile range, however throughout the document you also use mean and standard deviation, why both? You should also refer to whether the sample was normally distributed or not, in case it was not, what I suppose happened, why did you use parametric t test and anovas?! Please specify which intraclass correlation coefficient was used and where are these results. Furthermore, shapiro-wilk test is normally used for the identification of normality in a small sample, so why do you mention this test in table 1 and why do you refer to it as a mean to calculate homogeneity in distribution?

In the results section, line 157, please make it clear where is figure 1 in the document, I was not able to find it. Line 160-165, no need to repeat exactly the information that is in the table. Again, in table 1 you chose to use median and IQR, in table 2 you chose to use mean and SD, please try to be more consistent. Table 3, please add the comparisons between T1 and T2. To reduce the associated error, namely type 1 error you should perform all comparisons at once, with one statistical test, so please consider reviewing these results. Please add to this table the results for p<0.05, it is referred to in the document, but is not in the table.

Where is it possible to find in the document the results for the statistical comparison between the experimental and control groups? Please add this important analysis.

Please consider improving your discussion section, namely with more literature review. Line 195, the aims and objectives should be expressed always in the same way throughout the document. Line 205, “that changes in lower limb stability remained after the follow-up period”, is this a reference to your results? If so, how can you advocate this without measuring it? Line 207, I have never seen the measurement of a jump height as an instrument to evaluate co-contraction. This just does not seem plausible. Line 212, “with the absence of changes found in our study”, please make it clearer what did not change? Line 215, “for achieving improvement of lower limb stability in basketball players”, you can not say this based on your study sample, however, if this is a reference to another study, please add the reference.

Please enhance the conclusions section and add directions for future studies. Furthermore, make it clear that the present results are “suggestive of”, because it is not possible to conclude to the population.

Author Response

Reviewer 1

Comments and Suggestions for Authors

Dear authors, the paper was clear and well written, however some considerations are suggested:

  • Please consider revising your title. As suggested by the reviewer, we have amended the title of the article for improved understanding, according to the study design.
  • I do not think you have measured hamstring strength. As the reviewer points out, we have removed references to muscle strength. This variable was used in another study and was mistakenly included in this summary.
  • Make it clear what outcome measure is the "jumping" reference (eg., Performance, hight,...). We have included in the jumping reference that the measuring instrument relates to jump height.
  • The Y balance test has been referred as a test of dynamic postural control and stability, so perhaps you should elaborate a little bit more here and throughout the document. We have added more detailed information about the measuring instrument in the abstract and Instruments
  • Why don’t you refer to your study as a single-blind, randomized controlled pilot study? We have amended the title of the article for improved understanding, according to the study design.

Abstract section:

  • In the abstract section please consider changing the sentence on line 14 from “factors related to this injury” to “factors for this injury”. As pointed out by the reviewer, we have amended the sentence to improve understanding.
  • In the abstract you mention 17, but in the methods section you mention 14 subjects (7 for each group), please make it more clear. There was some confusion when indicating the number of subjects who completed the study. This has been corrected in the text and Figure 1 added (study flowchart), not initially included in the original submission.
  • In relation to your eccentric exercises, do you consider them to be calisthenics? If so, please add the information that no external weight was added. As noted by the reviewer, we have indicated that eccentric exercises were performed without external load.
  • You did not mention anywhere else in the document this “rupture angle” and again, I don’t really think you may say you have measured hamstring strength (line 18, 20, 23, 24). As the reviewer points out, we have removed references to muscle strength. This variable was used in another study and was mistakenly included in this summary.
  • In the reference to the single-leg squat and lunge (line 16 and 17), please make it clear this is a plyometric training. As noted by the reviewer, we have adapted the sentence to improve interpretation.
  • In the conclusions section, in the abstract (line 24-26), It is not possible to conclude that one intervention is better than the other without comparing them statistically and I do not think you have added these results to the document. When performing the analysis of repeated measures, we observed changes between the different evaluations in all dependent variables (intra-subject effect). However, there was no group interaction in any variable, so no intervention has been shown to be more effective than the other for improving the study variables.

Introduction section

  • Please consider adding “furthermore” before overuse in line 33. We have added “furthermore” accordingly.
  • Please revise the English language and style for the sentence “have contributed few provide evidence considered to be sufficiently effective” on line 35/36. We have revised the text so it is more easily understandable.
  • Line 38-40, in the sentence “The main etiological…. Flexibility” please consider the following “The main etiological factors for the development of hamstring injuries are poor hamstring-quadriceps ratio, the recurrence of previous injuries…”, please make it clear what you mean by “fatigue in the stages of greater muscle exhaustion”. We have inserted the corrections noted by the reviewer. At the end of the sentence we have added muscle fatigue as cause, adapting the sentence.
  • Line 46, in the sentence “the aim of this study is to verify the effectiveness of plyometric and eccentric exercises in improving the lower limb stability…” please consider the following “The aim of this study is to verify the effectiveness of a plyometric training program combined with eccentric exercises when compared to eccentric training alone in ….” As the reviewer points out, we have changed the text of study objectives to clarify the content of the paragraph.

Material and methods section

  • Please organize the materials and methods section in subheadings, namely ethical approval, participants, instruments, experimental design, and statistical analysis. We have divided the Material and Methods section using subheadings as noted by the reviewer.
  • Line 50, why don’t you add controlled after the single-blind statement? As suggested by the reviewer, we have included the word controlled in the design of the study.
  • Line 70, please remove the word “objectives”. The word has been deleted.
  • Line 77, please make a reference to the outcome measure related to the word "jumping". An error in the text that has been corrected. Jumping and lower limb stability were the dependent variables (not the measuring instruments, which are listed below). As the reviewer has pointed out, this sentence has been corrected to avoid confusion for the reader.
  • Line 97, why did you calculate interobserver reliability? Did both the rater and the researcher collect the data? And where are these results and the results for the intraobserver analysis, that is perhaps even more important? Before performing the pretreatment evaluation, we performed the pilot test to calculate the interobserver reliability between the principal investigator of the study and the blinded evaluator, to ensure proper performance of the evaluation protocol (described in the Instruments subheadings). Interobserver reliability was calculated using the intraclass correlation coefficient (indicated in the first paragraph of the Statistical analysis subheadings). The interobserver reliability results can be found in the first paragraph of the Results To facilitate the understanding and order of the contents, we have included this after the explanation provided in the subheadings Instruments.
  • Line 140-143, You refer to the use of the median and interquartile range, however throughout the document you also use mean and standard deviation, why both? Sample description and all other values have been shown with the same descriptors (mean and standard deviation).
  • You should also refer to whether the sample was normally distributed or not, in case it was not, what I suppose happened, why did you use parametric t test and anovas?! The T-student and ANOVA tests (parametric tests) were used after verifying the normal distribution of the sample in both groups at baseline.
  • Please specify which intraclass correlation coefficient was used and where are these results. The intraclass correlation coefficient was used to measure interobserver reliability in the pilot study developed prior to pre-treatment evaluation (as detailed in the preceding point).
  • Furthermore, shapiro-wilk test is normally used for the identification of normality in a small sample, so why do you mention this test in table 1 and why do you refer to it as a mean to calculate homogeneity in distribution? With the Shapiro-Wilk test we calculated the homogeneity of both groups at the beginning of the study, since the sample is less than 30 subjects (with a larger sample we would have used the Kolmogorov-Smirnov test). Table 1 shows the homogeneity with respect to the independent variables, and we have included the results of the analysis in the dependent variables at the beginning of the study.

Results section

  • Line 157, please make it clear where is figure 1 in the document, I was not able to find it. We have added Figure 1 (flow chart) which we mistakenly omitted in the original submission.
  • Line 160-165, no need to repeat exactly the information that is in the table. Again, in table 1 you chose to use median and IQR, in table 2 you chose to use mean and SD, please try to be more consistent. Following the reviewer's recommendation, we have removed the sample description. All details are in Table 1. Sample description and all other values have been shown with the same descriptors (mean and standard deviation).
  • Table 3, please add the comparisons between T1 and T2. As the reviewer points out, we have included the significance results in Table 3.
  • To reduce the associated error, namely type 1 error you should perform all comparisons at once, with one statistical test, so please consider reviewing these results. Please add to this table the results for p<0.05, it is referred to in the document, but is not in the table. In fact, to eliminate type I error, we performed the repeated measures analysis (Table 4) that we have included in this review. Similarly, we have added the pairwise comparison analysis (Table 5).
  • Where is it possible to find in the document the results for the statistical comparison between the experimental and control groups? Please add this important analysis. The group comparison analysis was performed with the repeated measures ANOVA, calculating Inter and intragroup differences (Table 4) and the pairwise comparison between T0-T1, T1-T2 and T0-T2 (Table 5).

Discussion section:

  • Please consider improving your discussion section, namely with more literature review. We have included a further review of the results and literature in the discussion section.
  • Line 195, the aims and objectives should be expressed always in the same way throughout the document. As suggested by another reviewer, we have removed the study objectives from the beginning of that paragraph.
  • Line 205, “that changes in lower limb stability remained after the follow-up period”, is this a reference to your results? If so, how can you advocate this without measuring it? We have amended the discussion section almost entirely, based on the comments of the three reviewers, to facilitate understanding.
  • Line 207, I have never seen the measurement of a jump height as an instrument to evaluate co-contraction. This just does not seem plausible. We have amended the Discussion section almost entirely, based on the comments of the three reviewers, to facilitate understanding.
  • Line 212, “with the absence of changes found in our study”, please make it clearer what did not change? We have reviewed the results and their interpretation in the Discussion
  • Line 215, “for achieving improvement of lower limb stability in basketball players”, you can not say this based on your study sample, however, if this is a reference to another study, please add the reference. We have redrafted the sentence to make it easier for the reader to understand.

Conclusions section:

  • Please enhance the conclusions section and add directions for future studies. Furthermore, make it clear that the present results are “suggestive of”, because it is not possible to conclude to the population. We have entirely changed the Conclusions section.

Reviewer 2 Report

Effectiveness of plyometric and eccentric exercise for hamstring strength, jumping and stability in female soccer players. A randomized pilot study

Introduction: Grammatical and presentation errors- can be improved

Good concise introduction but can talk more about evidence of type of exercise in Hamstring rehab especially type of eccentric exercise or plyometric.

Talk about reasons for choosing particular type of exercises in the experimental and control groups

Ln 30: re-phrase the line. I assume the authors want to say absence from sport. There is no” leave” from sport

Ln 31: consider using the word body region” instead of “body area”

Material and Methods:  Randomisation groups is not discrete and both groups have a common intervention. This will make it difficult to derive to any conclusion on which of the two groups did better

Ln 50-60: Write in past tense and a narrative text

Ln 50:  use narrative text and not a bullet point like statement

Ln 79-86: Re phrase the paragraph

Results:  Good representation with tables

Ln153-178: Grammatical errors, better choice of English narration will help the readers to have continued interest and involvement

Discussion:

Ln195-198. Repetition of the aim of the study, Rewrite the opening paragraph and start with the outcome of the study and not the aim.

The first paragraph reads that both interventions were useful—so what’s the significance of doing the study

Ln 199-206: Again, start with what this study found and your outcomes and not literature review

Ln 207-212 Again first talk about your study and then compare with the literature

Ln229: this randomised trial didn’t prove anything as both groups have similar interventions and hence the results on improvement in both groups

Conclusion: Unclear if the study answered the hypothesis

For argument, why would one add plyometric exercise to the eccentric ex, if the results are not significantly different

Topic of the paper reads about 3 outcomes: hamstring length, jumping and stability but the conclusion reads only about one

Conclusion: needs to be revised

Summary:

The paper involves a study on an important topic. The intervention group and experimental group didn’t explain if one intervention is better than the other. The study unfortunately doesn’t add much to the existing literature

Author Response

Comments and Suggestions for Authors

Effectiveness of plyometric and eccentric exercise for hamstring strength, jumping and stability in female soccer players. A randomized pilot study.

Introduction:

  • Grammatical and presentation errors- can be improved. As suggested by the reviewer, the text has been revised by a professional translator.
  • Good concise introduction but can talk more about evidence of type of exercise in Hamstring rehab especially type of eccentric exercise or plyometric. As suggested by the reviewer, we have included a further review in the introduction section.
  • Talk about reasons for choosing particular type of exercises in the experimental and control groups. As suggested by the reviewer, by including new text, we justify the use of both interventions.
  • Ln 30: re-phrase the line. I assume the authors want to say absence from sport. There is no” leave” from sport. As noted by the reviewer, we have made such amendments.
  • Ln 31: consider using the word body region” instead of “body area”. As noted by the reviewer, we have changed area for region.

Material and Methods

  • Randomisation groups is not discrete and both groups have a common intervention. This will make it difficult to derive to any conclusion on which of the two groups did better. The study groups were homogeneous at the beginning of the study and a common intervention was performed in both groups (eccentric exercises). The study hypothesis was whether the additional performance of plyometric exercises (experimental group) improved stability and jumping compared to the sole performance of eccentric exercises. With this quasi-experimental design, the objective was to evaluate the isolated and combined effectiveness of eccentric and plyometric exercises.
  • Ln 50-60: Write in past tense and a narrative text. As the reviewer indicates, we have revised the text, which has been corrected by a professional translator.
  • Ln 50: use narrative text and not a bullet point like statement. As the reviewer indicates, we have revised the text, which has been corrected by a professional translator.
  • Ln 79-86: Re phrase the paragraph. As noted by the reviewer, we have changed the sentence to make it easier to understand.

Results:  Good representation with tables.

  • Ln153-178: Grammatical errors, better choice of English narration will help the readers to have continued interest and involvement. As the reviewer indicates, we have revised the text, which has been corrected by a professional translator.

Discussion:

  • Ln195-198. Repetition of the aim of the study, Rewrite the opening paragraph and start with the outcome of the study and not the aim. At suggested by the reviewer, we have removed the study objectives from the beginning of the paragraph, providing the results of the study in a clearer way and easier to interpret.
  • The first paragraph reads that both interventions were useful—so what’s the significance of doing the study. The hypothesis of the study was that the combination of plyometric and eccentric exercises was more effective in improving jumping and lower limb stability than performing eccentric exercises alone. According to our results, this hypothesis cannot be assumed, accepting the alternative hypothesis (both interventions are equally effective). This approach has been included in the text.
  • Ln 199-206: Again, start with what this study found and your outcomes and not literature review. As indicated by the reviewer, we have started the paragraphs of the Discussion section with the results of our study, then including the review of other studies.
  • Ln 207-212 Again first talk about your study and then compare with the literature. As indicated by the reviewer, we have started the paragraphs of the Discussion section with the results of our study, then including the review of other studies.
  • Ln229: this randomised trial didn’t prove anything as both groups have similar interventions and hence the results on improvement in both groups. The hypothesis of the study was that the combination of plyometric and eccentric exercises was more effective in improving jumping and lower limb stability than performing eccentric exercises alone. This approach has been included in the text.
  • Conclusion: Unclear if the study answered the hypothesis. This has been included in Discussion.
  • For argument, why would one add plyometric exercise to the eccentric ex, if the results are not significantly different. As noted in the new review, in the Discussion section, we found no differences between the two groups. As recommendations for future research, the implementation of external weights or loads in the performance of plyometric exercises is suggested to establish if the hypothesis of our study is fulfilled.
  • Topic of the paper reads about 3 outcomes: hamstring length, jumping and stability but the conclusion reads only about one. We have amended the Conclusions section entirely.
  • Conclusion: needs to be revised. We have amended the Conclusions section entirely.

Summary:

The paper involves a study on an important topic. The intervention group and experimental group didn’t explain if one intervention is better than the other. The study unfortunately doesn’t add much to the existing literatura. By incorporating the full analysis of repeated measures to compare the two groups, we establish how both interventions are equally effective in improving stability. As in any study, we start from a null hypothesis that is to be verified. In this case, the hypothesis could not be confirmed. However, based on this study we cannot establish the efficacy of both interventions for improving lower limb stability. Similarly, the implementation of weight or loads is recommended in the performance of plyometric exercises for the improvement of dependent variables.

Reviewer 3 Report

The results can be of interest for direct application. As you pointed the low sample size is the main limitation of your study.

It is maybe difficult to consider it for publication as the sample size is a limiting factor, could you propose to complete these data ?

It could be otherwise of interest to collect/report data on further potential injuries in the Following months for example, to assess the long term protective effect of the protocol.

Author Response

Comments and Suggestions for Authors

The results can be of interest for direct application. As you pointed the low sample size is the main limitation of your study. As the reviewer points out, sample size is the main limitation of the study. Therefore, the main measures of methodological quality (randomization, always blinding, intention to treat analysis, follow-up evaluation and calculation of the size of the effect) have been implemented, to compensate for this limitation.

It is maybe difficult to consider it for publication as the sample size is a limiting factor, could you propose to complete these data? We have implemented the results with the full analysis of repeated measures and pairwise comparison. Similarly, we have included the results of the effect size in the repeated measures analysis, which allows to assess the power of the results, regardless of the sample size. Finally, in the subsection Limitations of the study, the sample size has been recognized as the main drawback of the study.

It could be otherwise of interest to collect/report data on further potential injuries in the Following months for example, to assess the long term protective effect of the protocol. Although evaluation in injury prevention was not the objective of this study, this issue has been included in the Recommendations for future research section. A reference to the relationship between improved dynamic stability and Injury Prevention has been included in the Discussion section.

Round 2

Reviewer 1 Report

Dear authors, I think you did an excellent job in your revised manuscript. In my opinion, it is now ready to be accepted for publication. If it is possible, please consider reducing the references to first-person pronouns.

Author Response

Thank you very much for your help to improve the manuscript. The text has been checked by a translator to remove some sentences with first-person pronouns.

Reviewer 3 Report

The manuscript has been significantly improved and meets the expectations. Thank you for you very clear answers and adapated modifications.

Author Response

Thank you very much for your help to improve the manuscript.